# Temporal analysis of academic performance in higher education before, during and after COVID-19 confinement using artificial intelligence

Laia Subirats[1,2], Aina Palacios Corral[1], Sofía Pérez-Ruiz[3], Santi Fort[1], Gómez-Moñivas Sacha[3]*

1 Eurecat Academy, Eurecat—Centre Tecnol'ogic de Catalunya, Barcelona, Spain, 2 ADaS Lab, Universitat Oberta de Catalunya, Barcelona, Spain, 3 Department of Computer Engineering, Universidad Autónoma de Madrid, Madrid, Spain

* sacha.gomez@uam.es

**Data Availability Statement:** All relevant data are within the paper and its Supporting Information files.

## Abstract

This study provides the profiles and success predictions of students considering data before, during, and after the COVID-19 pandemic. Using a field experiment of 396 students and more than 7400 instances, we have analyzed students' performance considering the temporal distribution of autonomous learning during courses from 2016/2017 to 2020/2021. After applying unsupervised learning, results show 3 main profiles from the clusters obtained in the simulations: students who work continuously, those who do it in the last-minute, and those with a low performance in the whole autonomous learning. We have found that the highest success ratio is related to students that work in a continuous basis. However, last-minute working is not necessarily linked to failure. We have also found that students' marks can be predicted successfully taking into account the whole data sets. However, predictions are worse when removing data from the month before the final exam. These predictions are useful to prevent students' wrong learning strategies, and to detect malpractices such as copying. We have done all these analyses taking into account the effect of the COVID-19 pandemic, founding that students worked in a more continuous basis in the confinement. This effect was still present one year after. Finally, We have also included an analysis of the techniques that could be more effective to keep in a future non-pandemic scenario the good habits that were detected in the confinement.

## 1. Introduction

Educational data mining (EDM) has been applied in several initiatives [1] and the use of computers in education is a hot topic nowadays [2]. A great amount of Massive Open Online Courses (MOOCs) has appeared offering their content completely online [3,4], including resources such as filmed lectures, problem sets and forums where the community can interact. This change in the learning methodologies (and the massive use of digital information) led to

**Funding:** This study has been partially funded by ACCIO, Spain (Pla d'Actuaci´o de Centres Tecnol'ogics 2021) under the project TutorIA. This study was also funded by the Fondo Supera COVID-19 (Project: Development of tools for the assessment in higher education in the COVID-19 confinement). The funders had no role in study design, data collection and analysis, decision to publish, or preparation of the manuscript.

**Competing interests:** The authors have declared that no competing interests exist.

the appearance of huge amounts of data that can be analyzed to improve teaching and learning quality (Learning Analytics) [2,5,6].

Moreover, between March and April 2020, distance learning was widely used during the COVID-19 confinement [7,8] due to the limitations of face-to-face activities. In this scenario, the improvement of online learning strategies became a necessity even to be applied institutionally [9]. In this context, the term emergency remote learning (ERL) was coined [10]. Fortunately, due to the interest in distance education that already appeared years before the pandemic, many e-learning platforms were available and widely used during the confinement [11], including many advanced tools such as intelligent tutoring systems (ITS) [12].

It is evident that a rigorous analysis of all the elements influencing students' performance in distance learning has become a fundamental requirement to understand the scope of the pandemic in education. This analysis becomes even more relevant in the current situation, where the pandemic is beginning to be left behind and both the transitory influence that occurred during confinement and the effects that will be permanent can be evaluated. In that sense, we will analyze in the present article data from students during courses before the pandemic; from the course 2019/2020 when they were forced to be confined and study at home, and also from the course 2020/2021 where students were not completely confined anymore.

In general, one of the elements that were more affected by the confinement was the time distribution in students' autonomous learning [13]. In the year that they were forced to study at home (2019/2020), students worked in a more continuous way due to different factors. On one side, they had more free time since they were not allowed to even go out of their homes. On the other side, more activities were proposed by the teachers to replace the final face-to-face evaluation tests. In the academic year after the confinement (2020/2021), many of these methodologies changed again to the previous format. However, the question now is if the experience lived by the students and the new habits acquired during confinement could last over time.

The first objective of this article is to analyze students' profiles related to time distribution in their autonomous learning before, during and after the COVID-19 confinement. The goal of this first objective is to find out if the effects in learning, beneficial on many occasions, that occurred in confinement are maintained over time. The second objective is to determine if the data collected is adequate to predict students' performance in order to apply additional measures to improve their learning if needed. Finally, we discuss about the possible reasons of students' wrong learning strategies, based on the effect of the pandemic, and the potential solutions to keep over time the most adequate time distribution in their autonomous learning.

## 1.1 Background

It is well-known that students can apply several learning strategies, and many different approximations and definitions can be found in the literature. For example [14], discusses the three major types of theories in learning: quantitative, qualitative, and behaviorist. The first two learning theories can be explained by defining four elements: instruction, learning process, learning outcome and performance. However, in the behaviorist model, the learning process and outcome are ignored. Another approximation is the study of motivation in learning strategies. For example [15], suggests four sources of motivation: vocational, academic, personal and social.

Some studies have also analyzed the relationship between learning strategies and academic performance in distance education. For instance [16], found that time management and effort, as well as complex cognitive strategies, were positive predictors of academic performance, whereas contact with others had a negative effect.

There are different types of students with different learning strategies. There are some who study continuously and those who leave it for the last moment and procrastinate [17]. describes that procrastination is related with greater disorganization and less use of cognitive and meta-cognitive strategies. In addition [18], found in their study that procrastination was negatively related to their test performance.

This scenario became even more complex in the past few years, when the pandemic affected our lifestyle and left us in an uncertain time. Among the most important effects on peoples' life, many people changed their daily routine, had to stay locked in their house, modified their social life completely and, even today, had to deal with long-lasting or permanent impact from the COVID-19. In terms of education, students also replaced their face-to-face lessons with online courses, had to change their study plan and tried to deal with a completely new educational system. In education, as well as in life in general, some difficulties had to be overcome due to the confinement. Depression, anxiety, poor internet connectivity, and an unfavorable study environment at home are a few examples of a very complex situation. It has been also reported [19] that 70% of learners were involved, motivated and highlighted the necessity of providing resilient education. It has been also reported that the pandemic impacted higher education not only in online classes but also in libraries (closed), social life, personal financial situation, and emotional health [20]. However, some studies also pointed out that students' performance during COVID-19 improved [13,21,22]. In addition, there are some studies such as [23] that collected sociodemographic and health variables, completing a measure of positive changes.

Predicting the evolution in students' learning behaviour due to the effects of the past months has become a possibility and may help us to understand the impact of the pandemic. Some studies also tried to understand how some of these factors can modify learners' habits. For example [24], uses scores to measure the effect of learning changes in students. Detecting any temporal progress of students has been also a topic of interest [25]. In a more technical and numerical approximation [26], examined the temporal dynamics of self-regulated learning behaviours using the multilevel vector autoregression (VAR) model. In [27], the effects of COVID-19 in college students' lessons is also analyzed.

It is clear that this unstable period may have changed students' habits, which can reverberate at their marks. By using temporal data and historical information, it has been demonstrated that it is possible to predict the final grade, which can be useful to correct wrong learning strategies. For example [28], brings analysis articles related to this field. Artificial intelligence has also become a useful tool in this scenario to predict, using historical data, mid and final marks. For example, to predict academic grades and dropout [29,30], used classification [31–33], used regression in higher education and high school datasets, respectively. Moreover, students' marks may be predicted using clustering as [34] or applying supervised machine learning algorithms as [35,36]. In other articles, it has been pointed out that those approximations can be also good indicators to predict the future marks of a student in soft skills subjects [37].

Pandemic changed our lives, and in the future, we may have to deal with similar global situations. Now, we can rely on this data to help our future students in similar circumstances, or to learn from the experience and keep the good learning techniques used in this period. In this complex and unique scenario, the study proposed in this article, where a full analysis of students' time management including pre-pandemic, pandemic and post-pandemic data, can be of great help to understand the effects of the confinement and improve learning methodologies in the future.

## 1.2 Purpose

The aim of this study is to analyze students' performance using data analysis focused on time distribution in three scenarios: before, during and after the COVID-19 pandemic. After doing

so, we will extend the results by applying an Artificial Intelligence methodology for predicting students' performance. The prediction methods proposed here are unsupervised learning and regression algorithms considering the temporal nature of the data acquired during the last five years. Data has been collected by the E-valUAM web application [38], as we explain in the methodology section.

These objectives can be achieved by answering the following research questions:

1. Does the profile of students vary before, during, and after the COVID-19 pandemic? To which extent did it vary?

2. Can Artificial Intelligence predict students' learning performance? To which extent?

3. Which prediction methods are more effective to predict students' performance before, during and after COVID-19 confinement?

4. What are the possible reasons of the different students' time management strategies in the three periods?

5. How can we improve and maintain in the future the most effective students' strategies found in the three periods?

## 2. Materials and methods

### 2.1 Measurement instruments

The subject of the study was"Applied Computing". This subject was taught through theory lessons and practical classes in the computer laboratory. This course corresponds to 6 ECTS and belongs to the first course (17–19 years old students) of the Chemical Engineering Degree in the Faculty of Sciences from Universidad Aut´onoma de Madrid, Spain. The course starts in Mid-February and ends in Mid-May (three months approximately). Due to the COVID-19 pandemic, the face-to-face teaching was cancelled on March 11, having a strong impact on the 2019/2020 course since it started in the first week of February and lasted until mid-May. Therefore it should be considered that the first 3 courses were normal (prepandemic) ones, 2019/2020 was the year of the pandemic since classes were online most of the year (after March 9 in Spain, with the course staring in mid-February), and 2020/2021 was the year after COVID-19 confinement with a mixed methodology of face-to-face and virtual teaching.

The subject uses the platform e-valUAM as a support tool for distance learning. The platform has been used in all years under study. Teachers and teaching methodology did not change either (excepting the online teaching due to the pandemic). E-valUAM platform implements computer-adaptive tests (CAT) proposed by Lord in 1980 [39–41]. CAT allows personalizing the next question of the test considering student's last answers. In Fig 1 we show an example of the possible paths that students can face when using e-valUAM in the subject"Applied Computing". Right answers imply a horizontal movement right on the diagram, while wrong answers imply a vertical drop on the diagram. The lowest grade (grade 0) corresponds to a test where the student has failed all the questions. The highest grade (grade 3) implies that the student has faced questions from all the possible level values (1,2). It is worth noting that this figure is a simplification for clarity of the authentic test since they have 20 questions and 4 levels.

All data in this article comes from the e-valUAM adaptive tests, prepared specifically for the subject under study. Tests are composed of 20 questions that can be selected from a repository of 50, where all of them include numerical parameters that change every time a student uses the application. The questions are selected depending on the performance of the student in the

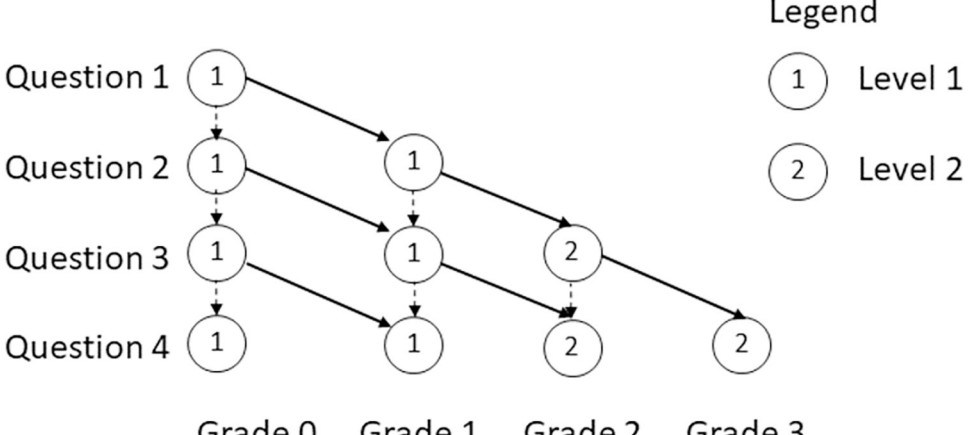

**Fig 1. Representation of the CAT model.** Solid and dotted arrows indicate the path of correct and incorrect answers respectively. Grades are represented at the bottom of the figure, In the Y axis there is the number of questions and in the circles there is the level of the question.

previous questions and, for that reason, the difficulty increases when students' performance increases. These kinds of tests are very useful for students in the sense that, when a student is stacked in a part of the subject, the tests give alternatives and more questions to improve that part. As soon as it is under control, the test goes to the next level.

Having numerical parameters that change all the time, students cannot learn the answer and must develop the whole procedure every time they are facing a question. Final exams have the same format and use the same application (e-valUAM). The tests for training through the application are available from the beginning of the semester.

An example of a first-level question would be:

"Give the solution of the following equation $Ax + B = (B + C)A$" where A, B and C are 4x4 matrices with numbers that change every time a student use the application.

An example of a last-level question would be:

"Give the solution of the following equation $3x + e^x + a = 0$, where $a = 3.17$". In this case, students should implement a numerical method explained in the theoretical lessons and determine the accuracy of the calculation. In this case, a is the parameter that changes every time a student faces the question.

It is worth noting that those questions are similar to the ones in the application. We are not showing the real ones for privacy reasons.

This teaching strategy has the following advantages:

1. Students are very motivated for using the web application and the tests because they know the final exam will follow the same format. This fact also implies that most of the time students are working autonomously they will be using the web application. Moreover, students need between 30 minutes and 1.5 hours to finish a test. This means that we do not have a significant delay between the moment they start a test and the end of the task.

2. Students need to follow the whole procedure every time they face a question

3. Since questions are selected randomly, students will face many troubles if they try to copy from a partner who is doing the test at the same time since it is not easy to be answering the same question at the same time (and even worse, numerical parameters will be different)

4. Since we are using the same application and test every year, factors such as different exams or more difficult ones do not apply.

## 2.2 Data description

For studying students' time management in their autonomous learning, we must decide first how and when we will be acquiring data. Data will be collected from e-valUAM platform as shown in the previous section along the whole academic year. The question is how to distribute these data in periods of time and which information is more important. Since we have many tests done by the students through the e-valUAM platform, we need first to filter data and group it in a few useful and handling group of parameters. Taking into account that the academic time for the subject under study is around 3 months, we have decided to group data in the following form:

- Mean mark of all the tests finished by a student in every month. These three parameters (one for each month) have values between 1 and 10. For simplicity, we will name them as marks1, marks2 and marks3, where marks1 corresponds to the first month of the subject and marks3 the month right before the final exams.

- Number of attempts by a student in every month. The three parameters (one for each month) have values between 1 and potentially infinite. In practical terms, it is usually below 100. For simplicity, we will name them as attempts1, attempts2 and attempts3 following the same distribution as explained in the previous point.

- Mark in the final exam. This is a single parameter (only final exam is counted) and it is only used for testing the algorithms since it is the value we want to predict. This parameter has values between 1 and 10. We will name this parameter as FinalMark.

As we mentioned before, the e-valUAM tests were available from the beginning of the academic year and students used them on a continuous basis. Although data is collected continuously, it comes from an adaptive test. This implies that there are some points in the test where students get stacked and some scores are repeated more often than others. For that reason, data are not following a normality distribution. As data did not pass the normality test, we used the non-parametric statistical test of Kruskal-Wallis [42] to compare the data and check statistical significance. The statistical analysis was performed using Python and the statistical significance was set at $p < 0.05$.

Raw data is composed of two different files or tables. The content of the first one is the e-valUAM data with the id of the student, the timestamp of the answer, and the mark. Indeed, students have several attempts since this test is their main tool in self-evaluation activities. There are also null attempts in the platform, which are attempts that are not finished and do not have a final mark. The second file is the mark of the final exam, so there are data of the student's id, the timestamp of the final exam, and the final mark.

For more information about the CAT model and e-valUAM platform refer to the methods section of the article [13].

## 2.3 Participants

We used data from all academic years between 2016/2017 and 2020/2021, composing a dataset of 396 students. Participants' data was collected in the following manner:

It does not involve minors.

- It has been collected anonymously. Students have been identified by a numerical code, avoiding gathering any personal information.

- Students have been informed by the lecturers that some information about their activity could be anonymously collected for statistical purposes. The authors of this study did not receive any objections.

- Tasks related to this study did not in any form alter students' activities, classes, or the assessment process.

Considering these circumstances, we have received confirmation from the Research Ethics Committee of Universidad Autonoma de Madrid that we do not need to apply for ethics approval from our university since no personal data, minors or potentially hazardous activities were involved in the study.

Teachers involved in the study (coauthors of the present manuscript) who were responsible for the subjects also gave consent to carry out the study. We obtained verbal consent from the participants in the study.

## 2.4 Numerical analysis

**2.4.1 Unsupervised learning algorithms.**   Unsupervised learning is a numerical methodology where algorithms find patterns on data sets and create clusters without using previous information.

Using unsupervised methods gives us the opportunity of getting groups of students that are classified without any previous information more than the time distribution oftheir activities in the subject since we will be using data collected and classified as shown in previous sections. In other words, by doing so, clusters will be only defined by students' time management. If we were using supervised learning algorithms, we had to link students' profiles to their final marks, which is something we need to avoid.

In this context, several algorithms can be applied such as k-means [43], hierarchical clustering [44] and Density-Based Spatial Clustering of Applications with Noise (DBScan) [45]. In this work k-means and hierarchical clustering are applied [46]. The first one is applied because it offers some advantages as it is relatively simple to implement, scales well, guarantees convergence, easily adapts to new examples and generalizes to clusters of different shapes and sizes, such as elliptical clusters (for more information see [47]). The second one is applied because is well suited for hierarchical data such as taxonomies, and dendrograms can be built.

In k-means, the elbow method [48]is used to find the optimal number of clusters. The method is a heuristic that consists of plotting the explained variation (in this case the within-cluster sum of square (WCSS)) as a function of the number of clusters, and picking the elbow of the curve as the number of clusters to use.

Before applying clustering, Principal Component Analysis (PCA) is performed to reduce dimensionality and the 4 attributes (mean number of months in the 3 months before the exam and the final mark) can be represented in a 3-D space. Performing PCA before clustering can alleviate the problem of the curse of dimensionality [49].

**2.4.2 Regression.**   The objective of the regression in this article is to predict students final marks in the subject from data extracted from their activities in autonomous learning, as explained in section 2.2.

Our goal is to find a regression model to predict marks accurately taking into account that our data is non-linear. For that reason, we have selected the MLPRegressor [50], a Multilayer Perceptor Regressor, a model based on neural networks, which performs very well with this kind of inputs [51].

Several experiments have been performed for regression:

1. Considering 30% of data randomly to test the regression

2. Not considering the last month data. This is specially useful to be able to warn students in advance.

3. Doing the prediction by years: using 2016/2017, 2017/2018 data to predict 2018/2019. This implies predicting a normal academic year from two normal academic years.

4. Doing the prediction by years: using 2016/2017, 2017/2018 data to predict 2019/2020. This implies predicting a confinement / virtual year from two normal academic years.

5. Doing the prediction by years: using 2016/2017, 2017/2018 data to predict 2020/2021. This implies predicting a mixed / semi-virtual year from two normal academic years.

For evaluating the performance of the regression, the Mean Absolute Error (MAE) has been used because it is one of the most used metrics in the literature (the lower the MAE the better) [29,52].

## 3. Results

### 3.1 Data statistics

In Table 1 we show a summary of the statistics in order to understand the nature of the data for each academic year. We could see that the academic year with the higher mean marks is 2020/2021, followed by 2019/2020. It should be noted that the threshold to pass the subject is 5.5 (instead of the usual 5).

Results show that the p-values were <0.05 thus statistically significant except for:

- 2016/2017-2018/2019 (both pre-pandemic)

- 2017/2018-2019/2020 (pre-pandemic and pandemic)

- 2017/2018-2020/2021 (pre-pandemic and post-pandemic)

- 2019/2020-2020/2021 (pandemic and post-pandemic)

There are not statistically significant between pre-pandemic years except 2017/2018, which is more similar to pandemic and post-pandemic. This means that we see statistical significance between pre-pandemic and pandemic/post-pandemic years if we exclude 2017/2018, which is a year where students got higher grades than those of pre-pandemic years exceptionally.

### 3.2 Unsupervised learning

Using unsupervised learning algorithms, we have found that three clusters can be obtained (details of the calculations are shown in the appendix.

In Table 2 we show data from the three types of student profiles using K-means:

1. "Continuous" which are the ones who obtain a high mark and study continuously during the semester,

2. "Last-Minute" students: those who study hard at the end of the semester

3. "Low-Perform" students, who have a low performance in the whole semester.

**Table 1. Description of the data.** SD stands for standard deviation.

| Final mark description | | | | |
|---|---|---|---|---|
| | **Mean mark** | **SD mark** | **# of fails** | **# of students** |
| Final evaluation 2016/2017 | 7.79 | 1.51 | 10 | 103 |
| Final evaluation 2017/2018 | 8.23 | 1.35 | 1 | 73 |
| Final evaluation 2018/2019 | 7.63 | 1.40 | 4 | 74 |
| Final evaluation 2019/2020 | 8.57 | 1.09 | 0 | 78 |
| Final evaluation 2020/2021 | 8.60 | 1.02 | 0 | 68 |

**Table 2. Description of the three clusters using KMeans.** Mean is used for the dataset attributes for each cluster.

| Clusters' description | | | | |
|---|---|---|---|---|
| | **Continuous** | **Last-Minute** | **Low-Perform** | **Total** |
| marks1 | 4.74 | 0.79 | 0.99 | - |
| attempts1 | 3.03 | 0.36 | 0.51 | - |
| marks2 | 6.9 | 0.64 | 1.0 | - |
| attempts2 | 3.28 | 0.44 | 0.31 | - |
| marks3 | 8.05 | 6.73 | 0.97 | - |
| attempts3 | 6.59 | 7.45 | 1.38 | - |
| FinalMark | 8.78 | 7.91 | 6.57 | - |
| 2016/2017 | 17 | 69 | 17 | 103 |
| 2017/2018 | 40 | 20 | 13 | 73 |
| 2018/2019 | 16 | 38 | 20 | 74 |
| 2019/2020 | 74 | 1 | 3 | 78 |
| 2020/2021 | 40 | 26 | 2 | 68 |
| Samplesize | 187 | 154 | 55 | 396 |

It can also be perceived that in 2019/2020 there were more"Continuous" students. That trend decreased in 2020/2021. However, continuous study levels in 2020/2021 have still higher values than before COVID-19 arrived.

In Table 3 we show data from the three types of student profiles using hierarchical clustering. It can be seen that clusters applying the hierarchical method (Table 3) are similar to the ones applying the K-means method (Table 2). This double check means that the clustering algorithms are showing robust results.

## 3.3 Regression

**3.3.1 All courses.** Before performing the predictions, the correlation heatmap of the attributes involved in the study is depicted in Fig 2. We can detect the following correlations:

**Table 3. Description of the three clusters using hierarchical clustering.** Mean is used for the dataset attributes for each cluster.

| Clusters' description | | | | |
|---|---|---|---|---|
| | **Continuous** | **Last-Minute** | **Low-Perform** | **Total** |
| marks1 | 4.2 | 0.88 | 0.98 | - |
| attempts1 | 2.7 | 0.36 | 0.53 | - |
| marks2 | 6.47 | 0.02 | 1.25 | - |
| attempts2 | 3.17 | 0.01 | 0.36 | - |
| marks3 | 7.91 | 6.47 | 0.36 | - |
| attempts3 | 7.1 | 6.6 | 0.73 | - |
| FinalMark | 8.73 | 7.69 | 6.63 | - |
| 2016/2017 | 29 | 60 | 14 | 103 |
| 2017/2018 | 44 | 21 | 8 | 73 |
| 2018/2019 | 21 | 35 | 18 | 74 |
| 2019/2020 | 75 | 0 | 3 | 78 |
| 2020/2021 | 45 | 21 | 2 | 68 |
| Samplesize | 214 | 137 | 45 | 396 |

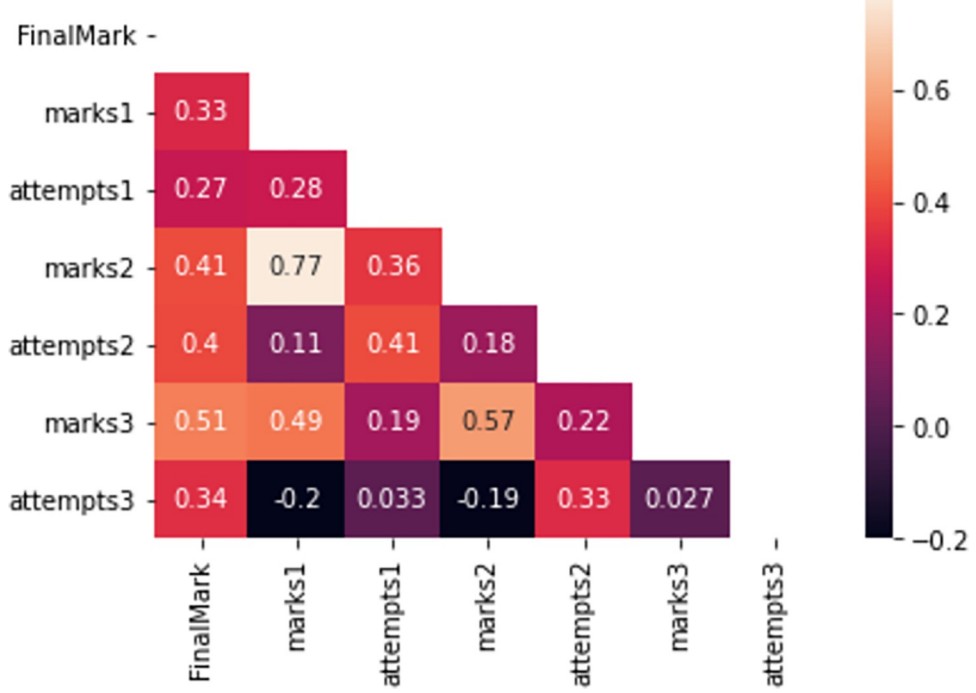

**Fig 2. Correlation heatmap.** Null attempts and the number of attempts are considered.

1. High positive correlation between marks1 and marks2, marks2 and marks3, and marks3 and FinalMark. This correlation is related to students who took good marks from the beginning of the semester and kept these good results until the end.

2. Negative correlations between marks1 and attempts3 and marks2 and attempts3. In this case, we are seeing the cases of students with good marks from the beginning who do not need to do a great effort at the end of the semester and do not use the application as much as they did at the beginning. This effect is higher in course 2019/2020, where they studied from home in the pandemic, working in a continuous basis more often than in the other periods.

For the prediction simulations, as explained in the methods section, data of all courses was considered taking 70% of data for training purposes and the rest 30% for testing and 5-fold cross-validation. The difference between the value and the prediction can be found in Fig 3. A threshold was established above 2. The value 2 indicates students that had marks 2 points better than expected. This could potentially be caused by malpractices such as copying. The threshold of 2 has been established because it is the 20% of difference of having a mark before the expected final exam. However, this threshold can be personalized by teachers according to their experience and subjects.

**3.3.2 All courses a month before the final exam.** In this experiment, as explained in the methods section, data of all courses was considered again taking 70% of data for training purposes and the rest 30% for testing and 5-fold cross-validation. However, in this case, e-valUAM data of the last month before the final exam was not considered. The purpose of this experiment is the predictions can be done with enough time to allow teachers to correct students' wrong learning strategies.

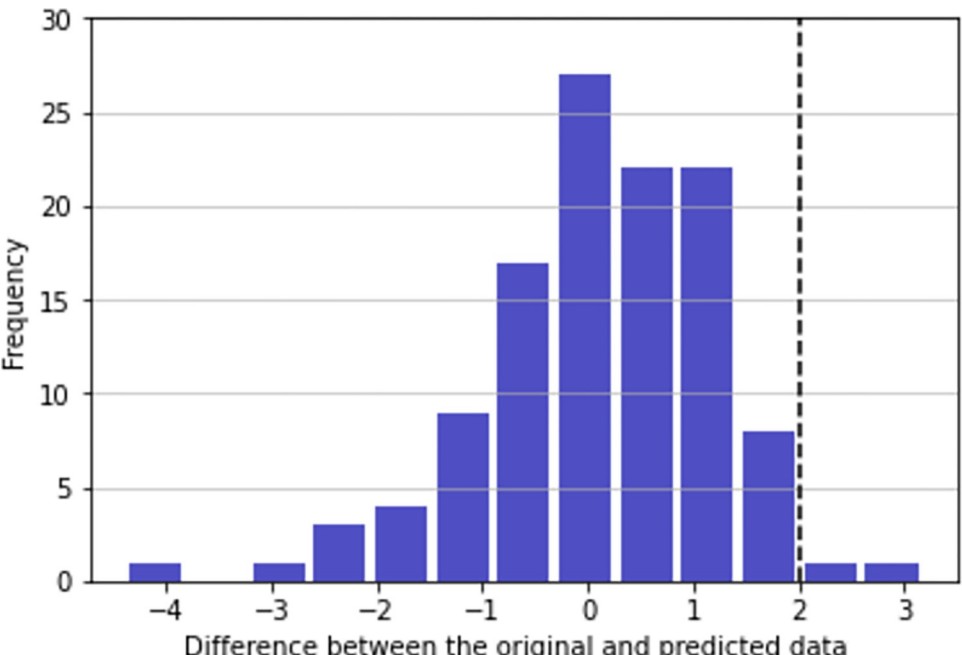

**Fig 3. Difference between the value and the prediction.** The dotted vertical line indicates a threshold from potential learning malpractice. Null attempts and the number of attempts are considered.

The difference between the value and the prediction can be found in Fig 4. In this case, the number of students with a risk of having a mark lower than 5.5 (the threshold in this subject) one month before the final exam is 4. As we can see in the figure, worse predictions appeared in this case, with error as high as 5 points over 10 in some particular cases.

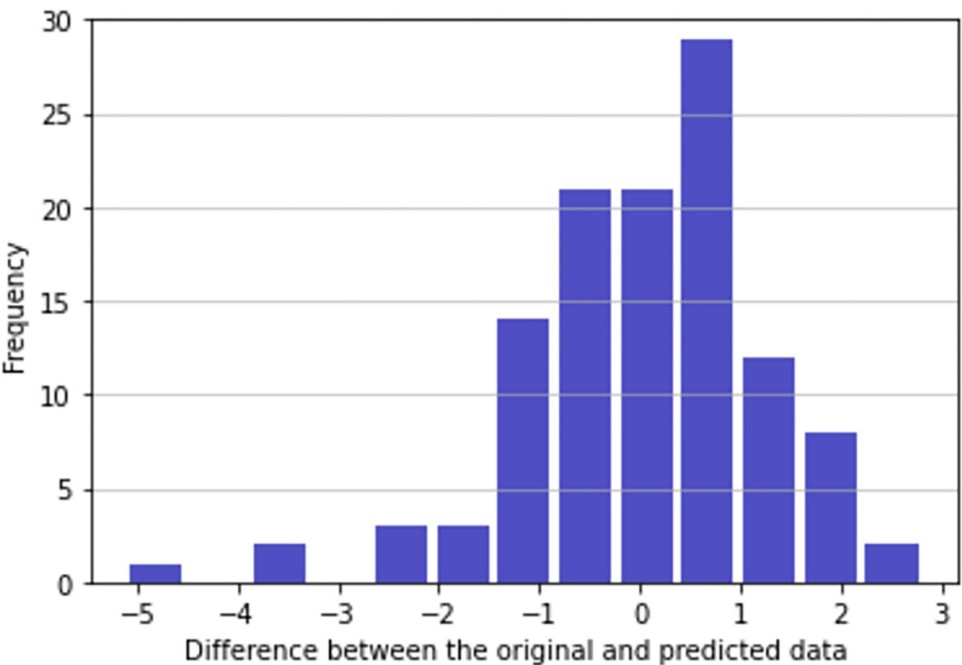

**Fig 4. Difference between the value and the prediction a month before the final exam.**

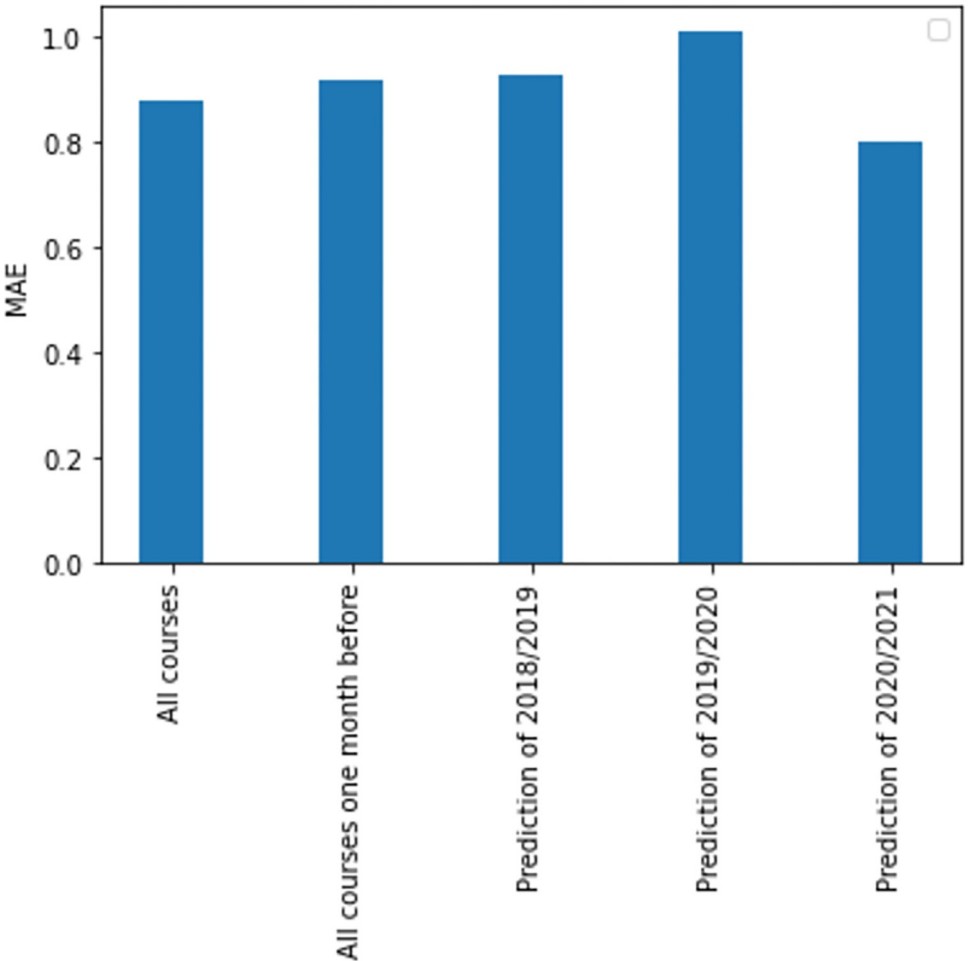

**Fig 5. Summary of MAE results in the experiments.**

**3.3.3 Prediction by years.**   In this experiment, as explained in the methods section, three different approaches were followed:

- Prediction of a course before the COVID-19 confinement: 2016/2017, 2017/2018 data to predict 2018/2019 (pre-pandemic year)

- Prediction of a course during the COVID-19 confinement: 2016/2017, 2017/2018 data to predict 2019/2020 (pandemic year)

- Prediction of a course after the COVID-19 confinement: 2016/2017, 2017/2018 data to predict 2020/2021 (post-pandemic year)

In Fig 5 a summary of MAE results is shown. We could see that the lowest MAE is the prediction of 2020/2021 (0.80), followed by all courses (0.88), all courses one month before (0.92), the prediction of 2018/2019 (0.93), and finally the prediction of 2019/2020 (1.01). Considering that the MAE is the average over the verification sample of the absolute values of the differences between forecast and the corresponding observation (forecast and corresponding observation have a value from 0 to 10), predictions are acceptable. We can find other MAE values in literature in [29,52].

## 4. Discussion

The main objective of this article is to analyze the different learning strategies of students and the influence of the distribution of their autonomous study time. This work is done taking into account the effect of confinement by COVID-19. Additionally, performance prediction strategies are proposed based on the analysis carried out.

In this section the five research questions proposed in the introduction will be answered:

1. Does the profile of students vary before, during, and after the COVID-19 pandemic? To which extent did it vary?

2. Can Artificial Intelligence predict students' learning performance? To which extent?

3. Which prediction methods are more effective to predict students' performance before, during and after COVID-19 confinement?

4. What are the possible reasons of the different students' time management strategies in the three periods?

5. How can we improve and maintain in the future the most effective students' strategies found in the three periods?

Answering research question 1, we have seen three different types of temporal profiles. We have observed that there is a group who studies continuously, another who studies at the end of the term, and the last one with very low performance in the continuous evaluation. The Low Performance group (who was not identified in previous works) has the highest risk of failure. Results from this point are useful in order to clearly identify the three types of students and to analyze the reasons why different groups have different performance. In that sense, we can distinguish success and failure. Success is related to Continuous and Last Minute groups and failure to Last Minute and Low Performance groups. The most interesting group is Last Minute. In this group, it is very difficult to predict success or failure since both concepts appear often. These results are in agreement with [53], where students who work continuously have a smaller chance of failure (defined as not passing the final exam). Although students who work mainly at the end of the term usually succeeded in their objectives, the risk of failure is much bigger than for Continuous group. The main reason could be that, in the case of having troubles in their learning process, studying continuously gives students the opportunity of searching for solutions. In the case of studying in the last minute, students do not have enough time.

Comparing the different courses of the subject under study, we can see that continuous students are more present 2019/2020, during the COVID-19 confinement. In the following year, this group was also present but with lower number than during the confinement (2020/2021). This information is very interesting since it demonstrates that some good habits (those which give better marks in the final exams) lasted one more 414 year at least. The problem now and a good question for future analysis is how long this 415 effect will be present. Another interesting fact is that students' marks were higher in 416 2020/2021 even when the continuous work started to decrease. There are some facts that could explain this apparent contradiction. First, in course 2020/2021 the restrictions were softer and teachers could use more materials and methodologies with students. Second, students could combine these better conditions with their (still) good habits in continuous working. A further study analyzing the methodologies used in the following years will be essential to confirm these arguments.

Regarding the second research question, the answer is yes since we have seen that, from a numerical point of view, using all data gives us very good predictions. We have even used two different clustering algorithms with similar results, which is a clear evidence of the robustness

of the methodology. However, removing data from the last month makes the predictions much worse. This means that we lost the most significant part of the information removing the last month. By analyzing the nature of the groups detected by our system, it is reasonable since Last Minute and Low Performance students cannot be distinguished without data from the last month (both groups performance was bad in the first two months). Since Last Minute students can still get good results and Low Performance ones cannot do it in general, the system fails when trying to predict the marks of these two groups.

The first conclusion here is that it is not easy to predict wrong learning strategies with enough time to apply measures to redirect students' habits into a more efficient way of working. We have tried to predict their performance with the whole data set (which includes data until the very last minute) and with data sets that only take into account the information obtained one month before the final exams. In this case, there are some students whom the system predicts are going to pass but in the end, they did not make the exam, which is a source of confusion to the algorithms. These mistakes are mainly due to the lack of information compared to the case when considering all data (it can be noted that the MAE is higher in this case). In general, students who study until the very last minute are taking the exam, so the source of error related to students who quit the subject is not present when considering the whole data set. However, since the predictions including all information are much more accurate, in this case, they could be useful to detect students' wrong learning strategies in a different sense. Regarding students that have a much higher score than expected, we could be detecting students that may have potentially copied.

In terms of students' perception, the existence of Last Minute students who succeeded is risky because the chances of success studying only the last few days can send a bad message and be tempting. Students could assume the risk of failure (hoping that they still have some chances of success) and leave most of the hard work to the last few days. It is important for the teachers to motivate students and explain well that it is much better for them to study continuously.

Answering research question 3, we have seen that regression algorithms described in the article are good predictors when all data are included. As usually happens in classification systems, the problem is more related to the quality of the data than to the performance of the algorithms. The worse results occurs always when we remove important sources of information such as students' data from their activities in the last month.

The second parameter that could have a strong influence is the academic years and their differences because of the pandemic. We can confirm, however, that the predictions are still effective in the years when the pandemic hit harder. Based on our simulations, we have seen that the best results are for the prediction of 2020/2021 considering 2016/2017 and 2017/2018 being all the other results also acceptable. Indeed, our algorithms performance is worse when not including the last month, but this fact happens independently of the academic year.

Regarding research question 4, we have seen that continuous working gives students a better chance of passing the final exams, as expected. This strategy was present much more often in the pandemic when they were studying at home. However, studying in the distance is not a guarantee by itself since there were many elements related to the pandemic that boosted this behavior. Some reasons could be that students were afraid of their future and did not have many other fun activities to do. Obviously, creating a similar scenario is not possible or even appropriate since it was a very dramatic experience for them. The main problem here is that, in a standard scenario such as the one previous to the pandemic, students are tempted to study in the last minute since they have seen many times that it is a strategy that could work, as we have demonstrated in the article.

Finally, answering research question 5, we have demonstrated that studying continuously in the pandemic improved students' performance. In that scenario, they did not need any

additional motivation since the pandemic itself created the perfect scenario for continuous working as we discussed in the answer of the previous question.

The problem now is how to lead students to the same good habits that they had in the pandemic without creating the dramatic situation of the pandemic. Obviously, the only way for improving continuous studying is to use additional motivational strategies. For example, using negative reinforcement such as the obligation of sending additional tasks along the semester or including some exams periodically could work. The problem with this strategy is that it has been used often before the pandemic and the results were not as good as they were in the pandemic. One of the reasons could be that punctual tasks are not really continuous working because students assume these midterm tasks as an exam and study again at the very last minute for each one of them. Assuming that students worked continuously in the pandemic by the reasons shown in answer to research question 4, the objective is to design a methodology where students have the subject in their mind continuously instead of using punctual tasks that would be more intermittent. Following this argument, using positive reinforcement techniques such as gamification are a good option because games are very attractive to students and they could have them in their mind for a longer and more continuous time.

## 5. Conclusion

This study shows the usefulness of determining profiles of students via clustering and regression. The clustering analysis detects three types of students: the ones who study continuously, the ones who study at the end of the term, and the ones with very low performance in the continuous evaluation. Students that work continuously have the best chance of success. Students in the group with low performance usually fail, and students who work harder at the end of the semester gave all kinds of results, making this practice risky. Knowing the type of students is a good help for teachers since they can detect and correct wrong learning strategies. On the other side, students can use this feedback to try different methodologies when they see that their path is not having a good ending.

Our results also show that it is possible to predict the final marks of the students when considering data from the beginning of the semester to the last day before the final exam. However, when trying to predict a month before the final exam, the error increments (an acceptable prediction is still obtained in some cases). Predicting a month before the final exam is useful to warn students and correct wrong learning strategies. However, in this case, we cannot avoid having some errors, mainly due to the removal of very important data related to students' activity at the end of the semester.

For this reason, some students at risk cannot be detected with enough time to apply additional activities or methodologies to correct their strategies. The use of predictions from the whole dataset is still very helpful to detect malpractices such as copying. It can be highlighted also that predictions are working well before, during and after the confinement due to COVID-19, which was not evident since students were forced to follow different learning methodologies.

Our study clearly indicates that students worked in a more continuous basis in the pandemic, and this behavior continued right after the pandemic. Since the continuous study has been demonstrated to be the most effective strategy found in our analysis, as expected, the goal now is to maintain this learning strategy in the future. We propose the use of gamification strategies since they are very motivating and can be used in a continuous basis.

For future work, this methodology could be applied to other subjects, universities and countries. It would be particularly interesting to perform a study with students with accessibility problems to the Internet or that have technical limitations in order to tackle Sustainable

Development Goal (SDG) 4—Quality Education (https://www.un.org/sustainabledevelopment/education). The study in different scenarios will be helpful to detect what events are occurring most often in some specific subjects or educational systems. It is worth noting that we have concluded that the different learning strategies applied before and after the confinement, as well as the learning conditions when working at home, were very relevant to change students' habits to a more continuous study. This fact could be altered when applying in a different country with different restrictions. Another future direction would be to extrapolate this work to subjects that include soft skills (such as arts and language subjects). These subjects can have qualitative information that may enrich the system but may impose a challenge to tackle this type of subjective information. A possible solution would be to apply sentiment analysis algorithms (there are some libraries that extract the polarity and subjectivity of a sentence such as Textblob) in order to make this subjective information more measurable.

## 6. Appendix. Computational details

### 6.1 Data cleaning

All code was implemented in Python 3.8.3, and scikit-learn library [54] was used.

The first action before performing the machine learning algorithms has been the cleaning of the data and their transformation in order to include in each instance a student and temporal information. For doing so, a single dataset from the two above-mentioned tables is created with the following attributes:

- marks1 [0–10]: the mean mark of the month before the penultimate month

- attempts1: number of attempts of the month before the penultimate month

- marks2 [0–10]: the mean mark of the penultimate month

- attempts2: number of attempts of the penultimate month

- marks3 [0–10]: the mean mark of the last month

- attempts3: number of attempts of the last month

- FinalMark [0–10]: the final mark of the subject

This data cleaning has been done using the month obtained from the timestamp of the continuous evaluation of each student, and then grouping the information of every student in the same instance across the above-mentioned attributes. Earlier attempts from those mentioned above (before the penultimate month) were not considered because they are not available along all academic years.

Two approaches have been studied: considering null attempts and without considering them. In the first approach, null attempts are considered both in the mean mark of the month and in the count of attempts. In the second approach, null attempts are deleted. Depending on the purpose of the study, one approach or the other one will be used (depending on which one has a better result).

### 6.2 Regression implementation details

For regression, we have used MLPRegressor with the following parameters for the gridsearchcv:

grid params MLPRegressor = ['MLPRegressor solver': ['lbfgs'], 'MLPRegressor max iter': [100,200,300,500],

'MLPRegressor activation': ['relu','logistic','tanh'],

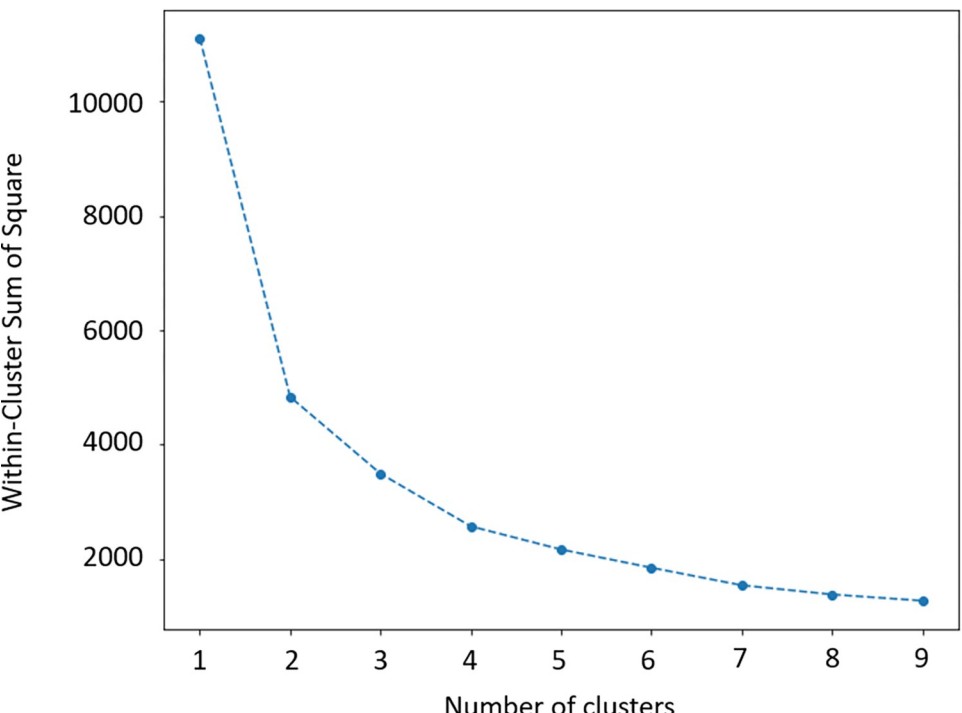

**Fig 6. Performance according to each number of clusters using unsupervised learning with k-means and PCA.**
Clustering without considering null attempts nor considering the number of attempts.

'MLPRegressor hidden layer sizes':[(2,), (4,),(2,2),(4,4),(4,2),(10,10),(2,2,2)],]

The StandardScaler was implemented in a Pipeline to standardize the data. This is particularly useful when there are different ranges of numbers in the attributes such as the mean of the mark and the number of attempts. Furthermore, 5-fold cross-validation has been applied.

After performing the experiments, the best parameters of the regression prediction have been 'MLPRegressor activation': 'relu', 'MLPRegressor hidden layer sizes': (4, 4), 'MLPRegressor max iter': 100, 'MLPRegressor solver': 'lbfgs'

Best parameters of a month before the final exam have been: 'MLPRegressor activation': 'logistic', 'MLPRegressor hidden layer sizes': (2,), 'MLPRegressor max iter': 100, 'MLPRegressor solver': 'lbfgs'

## 6.3 Clustering results

**6.3.1 k-means.** As explained in the methods section, we apply k-means, an unsupervised machine learning method which finds patterns in the data. Our goal is to divide the samples of each student by their behaviour during the grade. First, we apply PCA, an algorithm to reduce the dimension of the data, this algorithm also helps us to illustrate the samples in representable dimensions. K-means creates clusters with all the samples, but we need to find the optimal number of clusters, which represents the best division. For that, we use the elbow method which is found after computing the Within-Cluster Sum of Square (WCSS) of each cluster from 1 to 9 (see Fig 6). In Fig 6 we can see that as the number of clusters increases, the WCSS equals zero.

The optimal number of clusters can be found in the convex decreasing point of the graph, also called knee point. We used the knee locator library and inspecting Fig 6 visually, we can find that the elbow or knee locator is 3 (the best number of clusters is 3). After that, a

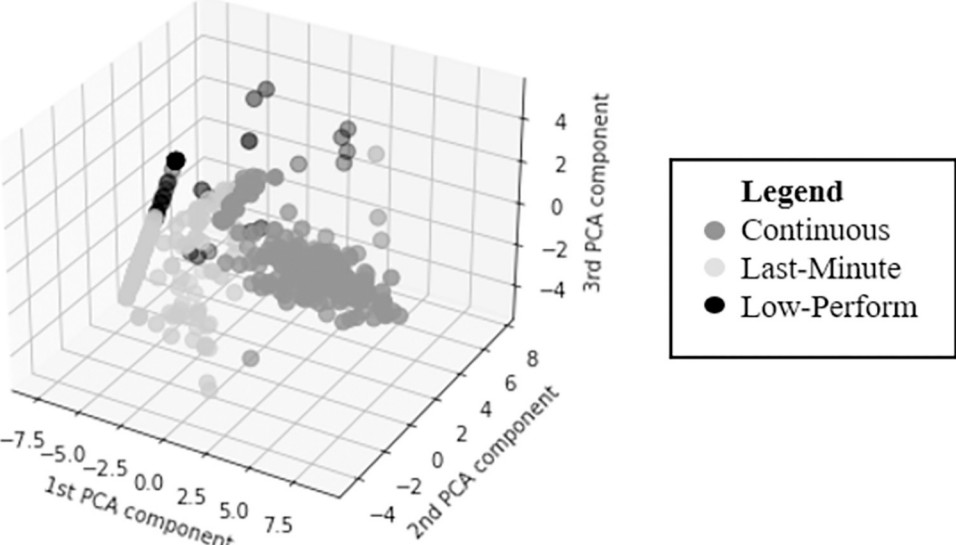

**Fig 7. User profiles using unsupervised learning with k-means and PCA.** Clustering without considering null attempts nor considering the number of attempts.

representation of the clusters is done in Fig 7. We can observe that there are in colors the 3 clusters (in black, grey and light grey) having as X, Y, and Z axis the 3 first components of the PCA. We can see that clusters are clearly differentiated and that low-perform ones are the least abundant cluster, followed by the last-minute ones. Finally, the continuous ones is the cluster with a bigger size.

In k-means, silhouette score is used to evaluate the clusters [55]. The best value if a silhouette score is 1 and the worst value is -1. Values near 0 indicate overlapping clusters while

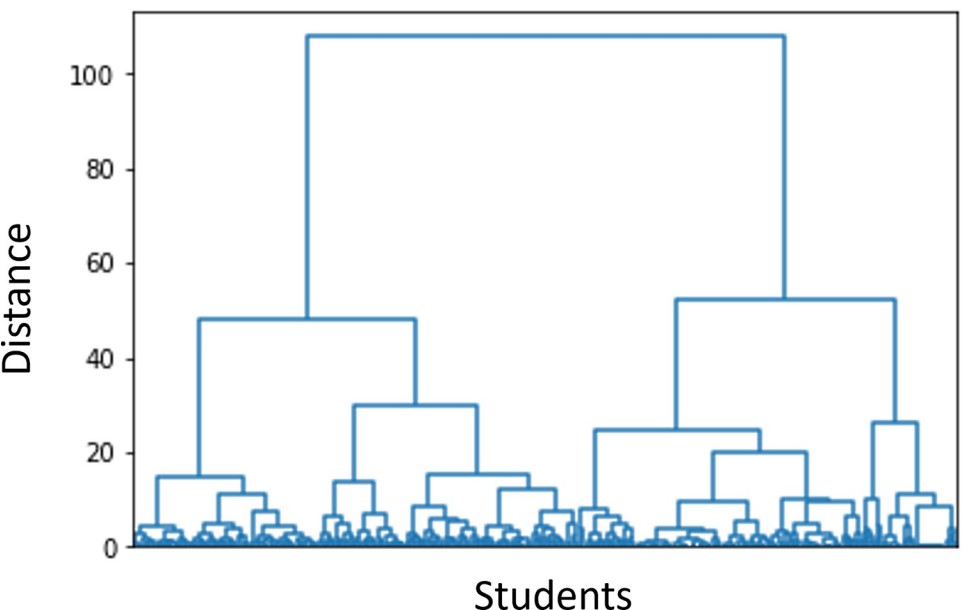

**Fig 8. Hierarchical clustering without considering null attempts nor considering the number of attempts.**

negative values generally indicate that a sample has been assigned to the wrong cluster because a more similar cluster can be found (for more information see scikit-learn).

Regarding the metrics of the cluster, the mean silhouette score is 0.48 and is higher in this experiment than considering null attempts nor the number of attempts. For this reason, the before-mentioned setup was conducted.

### 6.3.2 Hierarchical clustering

Regarding hierarchical clustering, the distance between clusters can be seen in Fig 8 where the following dendrogram is obtained applying hierarchical clustering. The dendrogram figure helps to see the distance between clusters and how they are separated.

## Supporting information

**S1 File. E-valUAM continuous evaluation data.** Grades of students during continuous evaluation of the subject"Applied Computing".
(CSV)

**S2 File. Final marks.** Final grade of students of the subject"Applied Computing".
(CSV)

## Author Contributions

**Conceptualization:** Laia Subirats, Go´mez-Mon˜ivas Sacha.

**Formal analysis:** Laia Subirats, Go´mez-Mon˜ivas Sacha.

**Investigation:** Laia Subirats, Aina Palacios Corral, Sof´ıa Pérez-Ruiz, Go´mez-Mon˜ivas Sacha.

**Methodology:** Laia Subirats, Aina Palacios Corral, Sof´ıa Pérez-Ruiz, Go´mez-Mon˜ivas Sacha.

**Project administration:** Santi Fort, Go´mez-Mon˜ivas Sacha.

**Resources:** Santi Fort, Go´mez-Mon˜ivas Sacha.

**Software:** Laia Subirats.

**Supervision:** Sof´ıa Pérez-Ruiz, Santi Fort, Go´mez-Mon˜ivas Sacha.

**Validation:** Aina Palacios Corral, Sof´ıa Pérez-Ruiz, Santi Fort.

**Writing – original draft:** Laia Subirats, Aina Palacios Corral, Sof´ıa Pérez-Ruiz, Santi Fort, Go´mez-Mon˜ivas Sacha.

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
