## [Decision Letter · Decision Letter 0]

29 Nov 2022

PONE-D-22-21370Temporal analysis of academic performance in higher education before, during and after COVID-19 confinement using artificial intelligencePLOS ONE

Dear Dr. Sacha,

Thank you for submitting your manuscript to PLOS ONE. After careful consideration, we feel that it has merit but does not fully meet PLOS ONE’s publication criteria as it currently stands. Therefore, we invite you to submit a revised version of the manuscript that addresses the points raised during the review process.

We look forward to receiving your revised manuscript.

Kind regards,

Qaisar Shaheen, Ph.D

Academic Editor

PLOS ONE

Journal Requirements:

“The funders had no role in study design, data collection and analysis, decision to

publish, or preparation of the manuscript.”

a)        Please provide an amended Funding Statement that declares *all* the funding or sources of support received during this specific study (whether external or internal to your organization) as detailed online in our guide for authors at http://journals.plos.org/plosone/s/submit-now. 

b)        Please state what role the funders took in the study.  If any authors received a salary from any of your funders, please state which authors and which funder. If the funders had no role, please state: "The funders had no role in study design, data collection and analysis, decision to publish, or preparation of the manuscript."

Please send your amended statements by return email; we will change the online submission form on your behalf.

Additional Editor Comments:

Prepare the draft as per the reviewer's comments

Reviewers' comments:

Reviewer's Responses to Questions

**Comments to the Author**

1. Is the manuscript technically sound, and do the data support the conclusions?

Reviewer #1: Yes

Reviewer #2: Yes

Reviewer #3: Partly

2. Has the statistical analysis been performed appropriately and rigorously? 

Reviewer #1: Yes

Reviewer #2: Yes

Reviewer #3: No

3. Have the authors made all data underlying the findings in their manuscript fully available?

Reviewer #1: Yes

Reviewer #2: Yes

Reviewer #3: Yes

4. Is the manuscript presented in an intelligible fashion and written in standard English?

Reviewer #1: Yes

Reviewer #2: Yes

Reviewer #3: Yes

5. Review Comments to the Author

Reviewer #1: This article studies the effect of COVID19 confinement in students’ habits by analyzing data of academic courses in three periods: pre-pandemic, confinement, and so-called post-pandemic. It also includes a methodology for predicting students’ performance by analyzing data of their activities and comparing with those of students from previous years.

The article is well-organized, and both the initial analysis and predictions are correctly linked and discussed in terms of the COVID19 influence in learning. The inclusion of data from three different periods gives the article a strong argument for validating the conclusions. The article includes several disciplines such as Education, Computer Science and, in a smaller percentage, Psychology. Authors use an appendix for including the details of computation, which is a good decision since it would be messy to include it in the main body of the article at the same time as details related to education and methodology.

The manuscript is well structured and presents interesting results. However, there are some points that need further consideration before suggested publication.

1)I think that research questions 2 and 3 could be combined in a single question. In fact, analyzing the extent of AI performance in question 2 it seems, it could be directly included in question 3. IN my opinion, If authors prefer to keep these two research questions, they should clarify the difference between them.

2)Do authors analyze the effect of team working in these three periods? It seems that working in groups is something strongly influenced by the pandemic since they cannot meet face-to-face. Does this factor have any influence in the results?

3)Paragraph starting in line 306 is hard to read. Maybe authors could rewrite it to make it clearer.

4)The dendrogram, in the appendix, is not correctly analyzed. It is not enough to say that “dendrogram figure helps to see the distance between clusters and how they are separated”. Authors should explain what they see in the dendrogram to justify their results.

Reviewer #2: The authors presented temporal analysis of academic performance in higher education before, during and

after COVID-19 confinement using artificial intelligence. The paper is well written and has potential for publication. I would recommend to accept it in this present form.

Reviewer #3: The title reflects what author intends to do clearly. The abstract presented is more subjective, inferred rather than including empirical findings discussed in the section "Results". The author claimed in abstract that "students’ wrong learning strategies", it is questionable? How can students achieved marks in exams reflect students' strategy for Exam preparation. The argument is not properly supported by the results discussion. The author added his opinion in an intuitive way in his introduction. For example, author says [During COVID] "2020), students worked in a more continuous way due to different factors. On one side, they had more free time since they were not allowed to even go out of their homes." How he knows that ? on the other hand he wrote "....... due to the confinement.Depression, anxiety, poor internet connectivity, and an unfavorable study environment at home are a few examples of a very complex situation".

The questions coined at page 3 are too common and none of those led to hypothesis whereas his study is of quantitative nature.

In Data Description section, the author says "data are not following a normality distribution........ we used the non-parametric statistical test of Kruskal-Wallis [42] to compare the data and check statistical significance." (see pge-6), Whereas at page 8, the author presents analysis in terms of means, STD etc. . The question is " Does statistical test of Kruskal-Wallis deals with means? The test he preferred to use is not followed in its true spirit which deal with ranks. So, how can results be justified?

The unsupervised learning section entitled clusters as "Continuous", "Last time", and "Low perform" were defined mentioning that a student" worked continuously", "study at end of semester" etc. not clear. How can these be inferred just from last results. Before clustering the author mentioned that " Principal Component Analysis (PCA) is performed to reduce dimensionality". What are variables involved in this process are not explicitly mentioned. Conclusion should be briefed for the reader. I feel the study is not technically too sound to be accepted for publication. My opinion may be subjective but what I inferred, I have documented it.

6. PLOS authors have the option to publish the peer review history of their article (what does this mean?). If published, this will include your full peer review and any attached files.

Reviewer #1: No

Reviewer #2: **Yes: **Dr. Muhammad Anwar

Reviewer #3: No

---

## [Author Response · Author response to Decision Letter 0]

24 Jan 2023

Reviewer #1: This article studies the effect of COVID19 confinement in students’ habits by analyzing data of academic courses in three periods: pre-pandemic, confinement, and so-called post-pandemic. It also includes a methodology for predicting students’ performance by analyzing data of their activities and comparing with those of students from previous years.

The article is well-organized, and both the initial analysis and predictions are correctly linked and discussed in terms of the COVID19 influence in learning. The inclusion of data from three different periods gives the article a strong argument for validating the conclusions. The article includes several disciplines such as Education, Computer Science and, in a smaller percentage, Psychology. Authors use an appendix for including the details of computation, which is a good decision since it would be messy to include it in the main body of the article at the same time as details related to education and methodology.

The manuscript is well structured and presents interesting results. However, there are some points that need further consideration before suggested publication.

1)I think that research questions 2 and 3 could be combined in a single question. In fact, analyzing the extent of AI performance in question 2 it seems, it could be directly included in question 3. IN my opinion, If authors prefer to keep these two research questions, they should clarify the difference between them.

Thank you for your suggestion. We combined research questions 2 and 3.

2)Do authors analyze the effect of team working in these three periods? It seems that working in groups is something strongly influenced by the pandemic since they cannot meet face-to-face. Does this factor have any influence in the results?

Thank you for the suggestion, we did not do it and it would be interesting. We have added a sentence related to this in the last section where future work is covered.

3)Paragraph starting in line 306 is hard to read. Maybe authors could rewrite it to make it clearer.

Thank you, we have rewritten it to make it more straightforward.

4)The dendrogram, in the appendix, is not correctly analyzed. It is not enough to say that “dendrogram figure helps to see the distance between clusters and how they are separated”. Authors should explain what they see in the dendrogram to justify their results.

Thank you, we have added a sentence to justify the distance between clusters.

Reviewer #2: The authors presented temporal analysis of academic performance in higher education before, during and after COVID-19 confinement using artificial intelligence. The paper is well written and has potential for publication. I would recommend to accept it in this present form.

Thank you.

Reviewer #3: The title reflects what author intends to do clearly. The abstract presented is more subjective, inferred rather than including empirical findings discussed in the section "Results". The author claimed in abstract that "students’ wrong learning strategies", it is questionable? How can students achieved marks in exams reflect students' strategy for Exam preparation. The argument is not properly supported by the results discussion. 

Thank you for this comment. We have revised the abstract and we have seen that we are missing some important points such as the origin of the data included in the 7400 instances. Without this information, the abstract is more difficult to read and understand. Indeed, reviewer is right when saying that, in the first version, some arguments are difficult to understand or believe. In the present version, we have clarified all the concepts and included more information to make the abstract clearer and better justified.

“This study provides the profiles and success predictions of students considering data before, during, and after the COVID-19 pandemic using a field experiment of 396 students. By processing more than 7400 instances with data from students’ activities taken from the whole academic course before the final exams, we have analyzed students’ performance considering the temporal distribution of autonomous learning. Our results show 3 main profiles: students who work continuously, those who do it in the last-minute, and those with a low performance in the whole autonomous learning. Although we have found that the highest success ratio is related to students that work in a continuous basis, last-minute working is not necessarily linked to failure. We have also found that students’ marks in the final exams can be predicted successfully taking into account the whole data of their activities in the academic course. However, predictions are worse when removing data from the activities of the month before the final exam. These predictions are useful to detect students’ learning strategies; prevent students from applying the less effective ones, and to detect malpractices such as copying. By analyzing data from courses 2016/2017 to 2020/2021, we have found that students worked in a more continuous basis in the confinement. This effect was still present one year after. Finally, We have also included a discussion of the techniques that could be more effective to keep in a future non-pandemic scenario the good habits that were detected in the confinement.”

The author added his opinion in an intuitive way in his introduction. For example, author says [During COVID] "2020), students worked in a more continuous way due to different factors. On one side, they had more free time since they were not allowed to even go out of their homes." How he knows that ? 

We agree with the reviewer that some sentences in the introduction are not well justified. In the present version, we have included new references to justify our arguments. For example, the fact the students had more free time is something that students told us during confinement. We did not include any reference about it for that reason. However, they were also interviewed by reporters and their answers were recorded in articles that are now included in the reference section.

on the other hand he wrote "....... due to the confinement.Depression, anxiety, poor internet connectivity, and an unfavorable study environment at home are a few examples of a very complex situation".

Actually, those negative factors were also present in the confinement and are compatible with the fact that students had more free time. Saying it with different words, students worked hard but they were not happy. In the introduction, we tried to included a complete background when we mentioned both positive and negative elements that had a strong influence in students’ learning performance. Again, we have justified this fact in the present version by including new references.

The questions coined at page 3 are too common and none of those led to hypothesis whereas his study is of quantitative nature.

We have reconsidered research questions to fit them to the quantitative nature of the article. Now, we have 4 research questions that are the product of combining research questions 2 and 3 in a single one. By doing this, we think that research questions are now easily to understand and fits better the purpose of our article.

In Data Description section, the author says "data are not following a normality distribution........ we used the non-parametric statistical test of Kruskal-Wallis [42] to compare the data and check statistical significance." (see pge-6), Whereas at page 8, the author presents analysis in terms of means, STD etc. . The question is " Does statistical test of Kruskal-Wallis deals with means? The test he preferred to use is not followed in its true spirit which deal with ranks. So, how can results be justified?

It has been added to the text that the Kruskal-Wallis deals with or compares the medians of independent samples. Both tests (Kruskal -Wallis and means, STD, etc.) are computed so both pieces of information are provided to the reader. The means, STD, etc. are easier to interpret, so, for this reason, it is justified that the clusters are named after this analysis.

The unsupervised learning section entitled clusters as "Continuous", "Last time", and "Low perform" were defined mentioning that a student" worked continuously", "study at end of semester" etc. not clear. How can these be inferred just from last results. 

We think that the main problem here is that we did not make clear the data we used in every case. For that reason, it is difficult to understand how we can obtain our results and conclusions. The following sentences have been used to clarify this point:

1) ”Continuous” which are the ones who obtain a high mark and study continuously during the semester (the mean in marks1, marks2 and marks3 is higher than the last-minute and low-perform clusters), 

2) ”Last-Minute” students: those who study hard at the end of the semester (the mean in marks1, marks2 and marks3 is low but the FinalMark is considerably high), 

3) ”Low-Perform” students, who have a low performance in the whole semester (the mean in marks1, marks2 and marks3 is low and they have the lowest FinalMark of the three clusters). 

Before clustering the author mentioned that " Principal Component Analysis (PCA) is performed to reduce dimensionality". What are variables involved in this process are not explicitly mentioned. 

It has been added to the text, that all 7 attributes depicted in Table 2 are involved in this process of PCA.

Conclusion should be briefed for the reader. 

We have carefully revised conclusion section and removed sentences that are not necessary or could be redundant. The following sentences have been removed to make it briefer and easier to understand:

● Our study clearly indicates that students worked in a more continuous basis in the pandemic, and this behavior continued right after the pandemic. Since the continuous study has been demonstrated to be the most effective strategy found in our analysis, as expected.

● such as arts and language subjects

● there are some libraries that extract the polarity and subjectivity of a sentence such as Textblob

I feel the study is not technically too sound to be accepted for publication. My opinion may be subjective but what I inferred, I have documented it.

We really appreciate reviewers’ comments and the detailed review of our article. 

We understand that some details were missed and the article could not sound as well as it should be. With the reviewers’ comments, and the article being modified accordingly, we hope it is adequate for publication now.

---

## [Decision Letter · Decision Letter 1]

14 Feb 2023

Temporal analysis of academic performance in higher education before, during and after COVID-19 confinement using artificial intelligence

PONE-D-22-21370R1

Dear Dr. Sacha,

We’re pleased to inform you that your manuscript has been judged scientifically suitable for publication and will be formally accepted for publication once it meets all outstanding technical requirements.

Kind regards,

Qaisar Shaheen, Ph.D

Academic Editor

PLOS ONE

Additional Editor Comments (optional):

Reviewers' comments:

Reviewer's Responses to Questions

**Comments to the Author**

1. If the authors have adequately addressed your comments raised in a previous round of review and you feel that this manuscript is now acceptable for publication, you may indicate that here to bypass the “Comments to the Author” section, enter your conflict of interest statement in the “Confidential to Editor” section, and submit your "Accept" recommendation.

Reviewer #1: All comments have been addressed

Reviewer #2: All comments have been addressed

2. Is the manuscript technically sound, and do the data support the conclusions?

Reviewer #1: Yes

Reviewer #2: Yes

3. Has the statistical analysis been performed appropriately and rigorously? 

Reviewer #1: Yes

Reviewer #2: Yes

4. Have the authors made all data underlying the findings in their manuscript fully available?

Reviewer #1: Yes

Reviewer #2: Yes

5. Is the manuscript presented in an intelligible fashion and written in standard English?

Reviewer #1: Yes

Reviewer #2: Yes

6. Review Comments to the Author

Reviewer #1: (No Response)

Reviewer #2: Comments have been addressed properly in the revised version of "Temporal analysis of academic performance in higher education before, during and after COVID-19 confinement using artificial intelligence"

7. PLOS authors have the option to publish the peer review history of their article (what does this mean?). If published, this will include your full peer review and any attached files.

Reviewer #1: No

Reviewer #2: No

---

## [Editor Report · Acceptance letter]

17 Feb 2023

PONE-D-22-21370R1 

Temporal analysis of academic performance in higher education before, during and after COVID-19 confinement using artificial intelligence 

Dear Dr. Sacha:

I'm pleased to inform you that your manuscript has been deemed suitable for publication in PLOS ONE. Congratulations! Your manuscript is now with our production department. 

Kind regards, 

on behalf of

Dr. Qaisar Shaheen 

Academic Editor

PLOS ONE